

# virMine: automated detection of viral sequences from complex metagenomic samples

Andrea Garretto[1], Thomas Hatzopoulos[2] and Catherine Putonti[1,2,3,4]

[1] Bioinformatics Program, Loyola University of Chicago, Chicago, IL, United States of America
[2] Department of Computer Science, Loyola University of Chicago, Chicago, IL, United States of America
[3] Department of Biology, Loyola University of Chicago, Chicago, IL, United States of America
[4] Department of Microbiology and Immunology, Loyola University of Chicago, Maywood, IL, United States of America

## ABSTRACT

Metagenomics has enabled sequencing of viral communities from a myriad of different environments. Viral metagenomic studies routinely uncover sequences with no recognizable homology to known coding regions or genomes. Nevertheless, complete viral genomes have been constructed directly from complex community metagenomes, often through tedious manual curation. To address this, we developed the software tool virMine to identify viral genomes from raw reads representative of viral or mixed (viral and bacterial) communities. virMine automates sequence read quality control, assembly, and annotation. Researchers can easily refine their search for a specific study system and/or feature(s) of interest. In contrast to other viral genome detection tools that often rely on the recognition of viral signature sequences, virMine is not restricted by the insufficient representation of viral diversity in public data repositories. Rather, viral genomes are identified through an iterative approach, first omitting non-viral sequences. Thus, both relatives of previously characterized viruses and novel species can be detected, including both eukaryotic viruses and bacteriophages. Here we present virMine and its analysis of synthetic communities as well as metagenomic data sets from three distinctly different environments: the gut microbiota, the urinary microbiota, and freshwater viromes. Several new viral genomes were identified and annotated, thus contributing to our understanding of viral genetic diversity in these three environments.

Corresponding author
Catherine Putonti, cputonti@luc.edu

## INTRODUCTION

In contrast to eukaryotic and prokaryotic organisms, only a small fraction of viral genomes has been sequenced and characterized. Viral metagenomic studies have been pivotal in increasing our understanding of viral diversity on Earth. Numerous habitats have been explored, such as: marine waters (*Breitbart et al., 2002*; *Yooseph et al., 2007*; *Hurwitz & Sullivan, 2013*; *Brum et al., 2015*; *Coutinho et al., 2017*; *Zeigler Allen et al., 2017*; see review *Brum & Sullivan, 2015*), soil (*Fierer et al., 2007*; *Zablocki et al., 2014*; *Adriaenssens et al., 2017*; see review *Pratama & Van Elsas, 2018*), freshwaters (*López-Bueno et al., 2009*; *López-Bueno et al., 2015*; *Roux et al., 2012*; see review *Bruder et al., 2016*), and the human

microbiota (e.g., *Reyes et al., 2010*; *Minot et al., 2011*; *Minot et al., 2013*; *Pride et al., 2012*; *Hannigan et al., 2015*; *Santiago-Rodriguez et al., 2015*; *Miller-Ensminger et al., 2018*; see review *Abeles & Pride, 2014*). Recent evidence has uncovered that viral members of the human microbiota (see reviews *Barr, 2017*; *Keen & Dantas, 2018*) and marine environment (see reviews *Breitbart et al., 2018*) play a more pivotal role than once thought. Regardless of the environment explored, the overwhelming majority of viral sequences produced exhibit no sequence homology to characterized viral species. Even for the well-studied marine viral communities, over 60% of the coding regions predicted are completely novel (*Coutinho et al., 2017*).

While metagenomics has been fruitful in identifying gene markers (e.g., 16S rRNA gene) and genomes of uncultivated eukaryotic and prokaryotic species (*Hug et al., 2016*), surveys of viromes face unique challenges (*Bruder et al., 2016*; *Rose et al., 2016*). First, unlike cellular organisms, there is no universally conserved gene in viruses. Viruses span a high degree of genetic diversity and are inherently mosaic (*Hatfull, 2008*). Second, even when sequencing purified virions, sequencing data often includes non-viral (host) DNA. This is further complicated by the fact that viral genomic DNA is often orders of magnitude less abundant than host cells or other organisms in the sample. In addition to the development of experimental procedures for viral metagenomics (e.g., *Conceição Neto et al., 2015*; *Hayes et al., 2017*; *Lewandowska et al., 2017*), several bioinformatic solutions have been created to aid in detecting viral sequences within mixed communities (e.g., *Roux et al., 2015*; *Hatzopoulos, Watkins & Putonti, 2016*; *Yamashita, Sekizuka & Kuroda, 2016*; *Ren et al., 2017*; *Amgarten et al., 2018*; see reviews *Hurwitz et al., 2018*; *Nooij et al., 2018*). Third, extant viral data repositories do not include sufficient representation of viral species. Thus, tools reliant upon identifying sequence homology, such as those for bacterial metagenome analysis (see review *Nayfach & Pollard, 2016*), have limited application in virome studies.

The identification of viral genomes from samples containing a single or a few viral species is relatively straight-forward, even in the presence of a large background of non-viral sequences. An example of such an inquiry would be the search for potential viral pathogens from clinical samples. Software tools including VIP (*Li et al., 2016b*), VirAmp (*Wan et al., 2015*), and VirFind (*Ho & Tzanetakis, 2014*) were designed specifically for such cases. They are, however, limited to the isolation of known viral taxa; complex viral communities pose significantly greater challenges. Typically, one of two approaches is taken. The first approach identifies contigs from metagenomic data sets based upon sequence attributes, e.g., their nucleotide usage profiles (*Ren et al., 2017*), and/or contig coverage (see reviews *Sharon & Banfield, 2013*; *Garza & Dutilh, 2015*; *Sangwan, Xia & Gilbert, 2016*). The second, more frequently pursued method, relies largely on recognizable homologies to known viral sequences, e.g., Phage Eco-Locator (*Aziz et al., 2011*), VIROME (*Wommack et al., 2012*), MetaVir (*Roux et al., 2014*), VirSorter (*Roux et al., 2015*), MetaPhinder (*Jurtz et al., 2016*), VirusSeeker (*Zhao et al., 2017*), and FastViromeExplorer (*Tithi et al., 2018*). The tool MARVEL integrates the two approaches, predicting tailed phage sequences based upon genomic features (gene density and strand shifts) and sequence homologies (*Amgarten et al., 2018*). Regardless of the approach taken, manual curation and inspection

is often a critical step in the process. Several complete viral genomes have been mined from metagenomic data through inspection of sequences based upon their size, coverage, circularity, or sequence homology to annotated viral genes or genes of interest (e.g., *Inskeep et al., 2013*; *Labonté & Suttle, 2013*; *Dutilh et al., 2014*; *Smits et al., 2014*; *Smits et al., 2015*; *Bellas, Anesio & Barker, 2015*; *Rosario et al., 2015*; *Zhang et al., 2015*; *Paez-Espino et al., 2016*; *Voorhies et al., 2016*; *Coutinho et al., 2017*; *Ghai et al., 2017*; *Watkins, Sible & Putonti, 2018*). These efforts have uncovered novel viral species, furthering our understanding of genetic diversity in nature.

Here we present virMine for the identification of viral genomes within metagenomic data sets. virMine automates the process of discovery; from raw sequence read quality control through assembly and annotation. virMine incorporates a variety of publicly available tools and user-defined criteria. In contrast to previous bioinformatic tools which search for viral "signatures" based on our limited knowledge of viral diversity on Earth, virMine takes advantage of the wealth of sequence data available for cellular organisms. Thus, viral (bacteriophage and eukaryotic virus) discovery is conducted through the process of excluding what we know not to be viral. Those sequences which are not "non-viral" (i.e., putative viral sequences) are then compared to a database of viral sequences. This comparison distinguishes putative viral sequences similar to known viral sequences and those which may represent novel viruses for downstream analyses. A beta version of this tool was used to isolate viral sequences from urinary metagenome data sets (*Garretto et al., 2018*). Here we illustrate the utility of this tool using four case-studies: synthetic data sets, gut microbiomes, urinary viromes, and freshwater viromes, resulting in the identification of new strains of known viruses as well as novel viral genomes.

## MATERIALS & METHODS

### Pipeline development

The pipeline integrates existing tools as well as new algorithms using Python and the BioPython library (*Cock et al., 2009*). Figure 1 depicts the process employed by virMine. A key aspect of the tool is its flexibility; it was designed to be modular, allowing users to access functionality individually or execute the full pipeline. While several methods have been incorporated in this release (Table 1), new tools can be added easily. Furthermore, to facilitate targeted analyses, filtration options and customization is available for users without any programming expertise.

Users can supply either raw Illumina sequencing reads (single-end or paired-end) or assembled contigs/scaffolds. In the case in which reads are supplied, the raw sequencing data is evaluated using the quality control tool Sickle (https://github.com/najoshi/sickle). Reads are trimmed, generating high quality data for assembly. Presently, the pipeline performs assembly by one of three methods: SPAdes (*Bankevich et al., 2012*), metaSPAdes (*Nurk et al., 2017*), and MEGAHIT (*Li et al., 2016a*). These assemblers were selected as they include tools better equipped for assembly of low complexity samples (SPAdes) and those developed for complex metagenomes (metaSPAdes and MEGAHIT). In a prior study comparing tools for assembly of phage genomes from single or low complexity samples

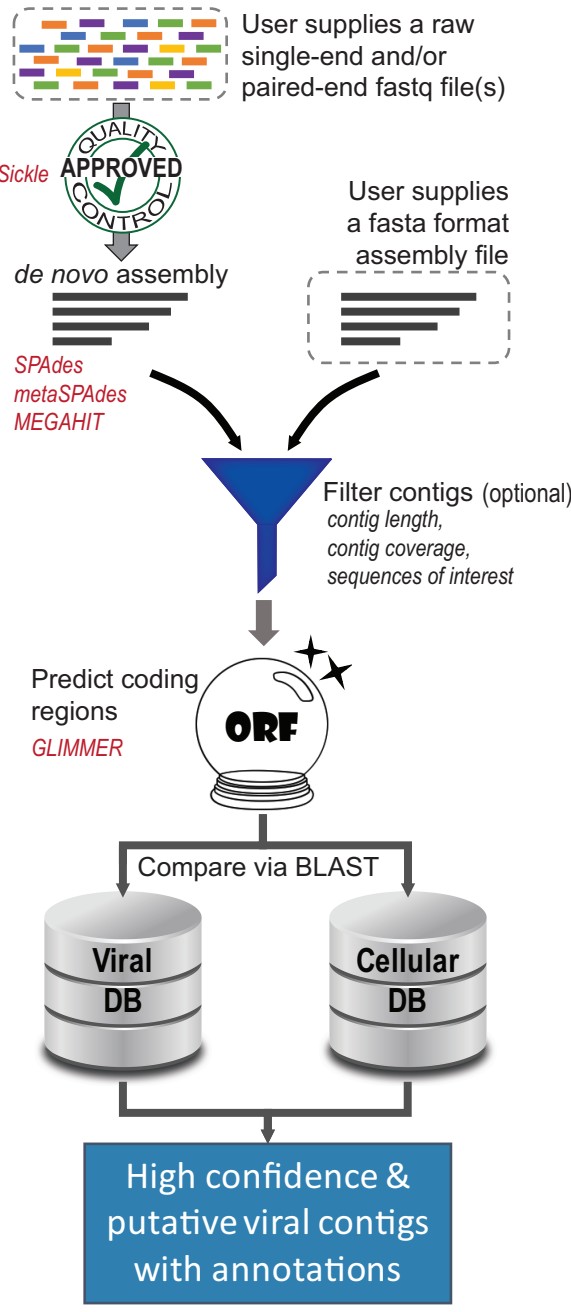

**Figure 1 Overview of virMine pipeline.** Tools integrated into the pipeline are listed in red. The sequences for viral contigs predicted with high confidence (''viral_contigs'') and putative viral contigs (''unkn_contigs'') are written to file.

(*Rihtman et al., 2016*), the SPAdes assembler (*Bankevich et al., 2012*) outperformed other tools tested. virMine also includes the assembly option ''all3''. This option assembles the reads using SPAdes, metaSPAdes, and MEGAHIT and selects the assembly with the highest $N_{50}$ score for downstream analysis. The virMine command line includes a flag for the

**Table 1  Software integrated into the virMine pipeline.**

| Tool | Version | Task | Citation |
|---|---|---|---|
| Sickle | 1.33 | Read trimming | https://github.com/najoshi/sickle |
| SPAdes | 3.10.1 | Assembly | *Bankevich et al. (2012)* |
| metaSPAdes | 3.10.1 | Assembly | *Nurk et al. (2017)* |
| MEGAHIT | 1.1.4 | Assembly | *Li et al. (2016a)* |
| BBMap | 37.36 | Coverage | https://sourceforge.net/projects/bbmap/ |
| GLIMMER | 3.02 | Gene prediction | *Delcher et al. (1999)* |
| BLAST+ | 2.6.0 | Sequence Analysis | ftp://ftp.ncbi.nlm.nih.gov/blast/executables/blast+/ |

user to specify the number of threads to be used during assembly to best utilize multi-core resources.

Next, virMine includes several options for the user to filter the assembled contigs. This can include minimum and/or maximum contig length, minimum contig coverage, and presence of genes or sequences (such as CRISPR spacer sequences) of interest. Coverage is calculated by remapping the original reads to the contigs, and the per contig coverage is calculated via BBMap (https://sourceforge.net/projects/bbmap/). Coverage is not reported if this option is not selected. Alternatively, when SPAdes (*Bankevich et al., 2012*) or metaSPAdes (*Nurk et al., 2017*) is used for assembly, users can select to use the SPAdes "cov" value as a filter. Users can also provide FASTA format sequences of interest (e.g., gene sequences encoding for a specific functionality); contigs are then queried against this data set using blastx. Results with a bitscore >50 are considered real hits and only contigs containing these hits will be considered further. Any or all of these filters can be selected by the user. Furthermore, the order in which they are specified by the user determines the order in which the filters are applied.

In Step 3, coding regions are predicted for each contig. Open reading frame (ORF) prediction is conducted using the tool GLIMMER (*Delcher et al., 1999*). Coding regions are predicted using a modified GLIMMER script (available through our GitHub repository), trained to accommodate characteristics of viral genes, e.g., overlapping genes (*Chirico, Vianelli & Belshaw, 2010*) and short coding regions.

In the final step, each predicted ORF is compared to two databases—a collection of non-viral sequences and a collection of known viral sequences. These two databases can be manually curated data collections or obtained from public repositories. While the GitHub repository for virMine includes a script to generate databases from NCBI's RefSeq collection, any multi-fasta file of amino acid sequences can be used to create these databases; the user need only supply the multi-fasta files. Comparisons against these two databases are facilitated via the BLAST+ application (*Camacho et al., 2009*). Users can select to use either a blastp (protein query) or blastx (translated nucleotide) query. While blastx is more exhaustive, blastp is more expedient. Again, the threads flag is used here to expedite these comparisons. All hits are reported from both databases; the bitscores for each ORF's hits to the two databases are compared, and the ORF is called "viral" or "non-viral" based upon the hit with the greater bitscore. Contigs with more "viral" calls are predicted as viral and are written to file ("viral_contigs.fasta"), as are their ORF predictions and BLAST

(either blastx or blastp) results. Contigs containing ORFs with no recognizable sequence homology to the viral database or non-viral database are classified as "unknown". These putative viral contigs ("unkn_contigs.fasta") and their ORF predictions are also written to file, as these sequences may represent truly novel species.

## Tool availability

virMine is available through a Docker image at https://github.com/thatzopoulos/virMine; Docker builds the necessary environment. This repository also includes scripts for generating curated viral and bacterial databases from GenBank. The user can save the contents of their run locally, as well as supply their own input files prior to the building of the environment, by following the steps listed in the GitHub repository. This pipeline can be run on any system supporting Docker (https://www.docker.com/). Development and testing were conducted on the Ubuntu and MacOSX operating systems.

## Data sets

The pipeline includes two databases for distinguishing between non-viral and viral sequences. Two data sets were created for our proof-of-concept work. The viral database includes amino acid sequences from all RefSeq (*O'Leary et al., 2016*) viral genomes and can be retrieved directly online at ftp://ftp.ncbi.nlm.nih.gov/genomes/Viruses/all.faa.tar.gz. This data set includes both eukaryotic viruses and phages. The non-viral data set used for our proof-of-concept work was created using the bacterial COGs collection (*Galperin et al., 2015*), excluding coding sequences in the category X of phage-derived proteins, transposases, and other mobilome components. The GitHub repository for virMine includes a script to create these two databases.

For the proof-of-concept studies presented in the results, four data sets were used. The first is a synthetic data set for benchmarking purposes. Sequencing read sets for a single "non-viral" sequence (*Pseudomonas aeruginosa* UW4 (NC_019670.1)) and a single viral sequence (*Pseudomonas phage* PB1 (NC_011810.1)) were created at various "concentrations" using the tool MetaSim (*Richter et al., 2008*). These synthetic data sets were made both with and without mutations introduced. (Mutations were introduced using the evolve function in which the parameters "number of leaves (Yule-Harding Tree)" and "Jukes-Cantor Model Alpha" were set to the defaults 100 and 0.0010, respectively.) Raw sequencing reads were also obtained from five different studies including the gut microbiota (*Qin et al., 2010*; *Reyes et al., 2010*), the urinary microbiota (*Santiago-Rodriguez et al., 2015*), and freshwater viromes (*Sible et al., 2015*; *Skvortsov et al., 2016*). Table 2 summarizes these data sets; details regarding the URLs for these data sets can be found in File S1.

Local BLAST searches of contigs were conducted using the complete nr/nt database (downloaded 6/24/2017). Remote BLAST queries were conducted through the NCBI website. Genome annotations were generated using RAST (*Aziz et al., 2008*), previously used for phage genome annotations (*McNair et al., 2018*). Contig mapping to complete genome sequences was performed using Bowtie2 (*Langmead & Salzberg, 2012*).

**Table 2 Complex community microbiomes examined for virMine proof-of-concept study.**

| Sample | Study details | Sequencing technology | # samples | # reads (millions) |
|---|---|---|---|---|
| Synthetic | *P. aeruginosa* and *Pseudomonas phage* PB1 genomes | N/A | 22 | 4.4 |
| Gut Microbiomes | A subset of faecal microbiota of monozygot twins and their mothers (*Reyes et al., 2010*) | 454 FLX | 3 | 0.66 |
| | A subset of faecal samples from 124 European individuals (*Qin et al., 2010*) | Illumina Genome Analyzer | 55 | 1141.33 |
| Urinary Viromes | UTI positive urine samples (*Santiago-Rodriguez et al., 2015*) | Ion Torrent PGM | 10 | 6.22 |
| Freshwater Viromes | A subset of samples from Lake Michigan nearshore waters (*Sible et al., 2015*) | Illumina MiSeq | 4 | 13.46 |
| | Viral community of Lough Neagh (*Skvortsov et al., 2016*) | Illumina MiSeq | 1 | 4.60 |

# RESULTS & DISCUSSION

virMine is a single tool to perform raw read quality control, assembly, annotation, and analysis (Fig. 1). The virMine tool, as described in the Methods, identifies viral sequences and putative viral sequences in metagenomic data sets by harnessing the wealth of non-viral sequence data available; contigs are scored based upon their similarity to non-viral and viral sequences. Four case studies were derived to test the efficacy of the virMine software tool, including one synthetic data set and three different environmental samples from the gut, urine, and freshwaters.

## Case study 1: synthetic data sets

Sequencing reads were generated using the tool MetaSim (*Richter et al., 2008*) using a sample "non-viral" genome sequence, *Pseudomonas aeruginosa* UW4 genome (GenBank: NC_019670), and a viral genome sequence, *Pseudomonas* phage PB1 (GenBank: NC_011810). Eleven synthetic data sets were created in which 0% through 100% (increments of 10%) of the data set comprised of "reads" from the phage genome sequence. Each synthetic data set was processed independently; assemblies were generated using SPAdes (*Bankevich et al., 2012*) with the requirement that the coverage (-cov flag) be greater than or equal to three.

Figure 2 summarizes the results of the analyses. When 50% or more of the reads were from the PB1 genome, the complete PB1 genome could be reconstructed. As the $N_{50}$ scores for each of the runs show, the length of the virMine assembled viral genome exceeds that of the PB1 genome (65,764 bps); this is a residual of the direct terminal repeats (DTRs) in the PB1 sequence. The presence of DTRs frequently leads to "wrap-around" reads contained within the genome assembly (*Merrill et al., 2016*). Each contig that did not correspond with the PB1 genome, including those identified within the 0% PB1 genome data set, was further examined via local blastn against the nr/nt database (File S2). As Fig. 2 shows, even for the synthetic data set with no reads from the PB1 genome, two contigs were predicted by virMine to be viral. We further investigated these contigs, 1021 and 1007 bp in length; the first contig is homologous to an IS3 family transposase (GenBank: AFY17357) and an IS110 family transposase (GenBank: AFY17680), respectively. As these transposases are
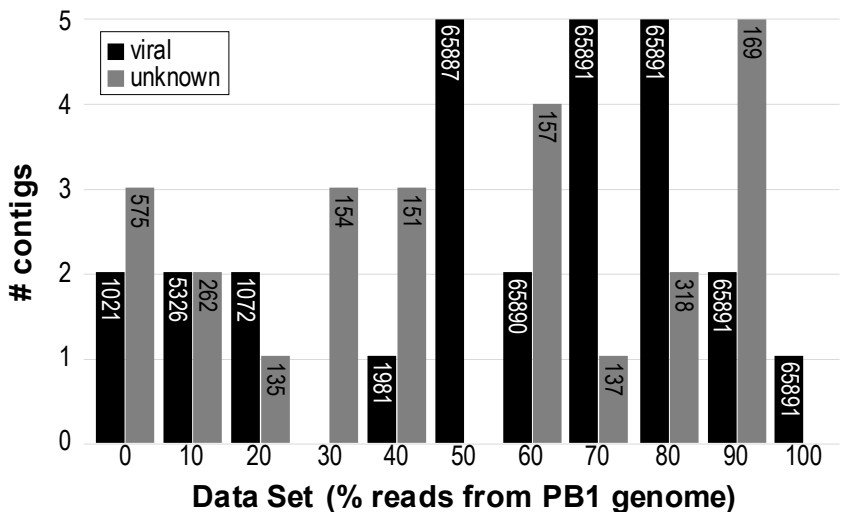

**Figure 2  Number of contigs assembled for each of the synthetic data sets predicted as viral (black bars) or of unknown origin (gray bars).** The $N_{50}$ score of the assembled contigs in each group is indicated within the corresponding bars.

assigned COG id numbers within the category X, they were excluded from the non-viral database and thus not recognized as non-viral. Transposases are abundant in nature and have been found within phage genomes (*Aziz, Breitbart & Edwards, 2010*).

MetaSim (*Richter et al., 2008*) was used again to produce synthetic data sets for the *P. aeruginosa* and *Pseudomonas* phage PB1, this time introducing mutation (population-based random mutator; see Methods). As shown in File S2, the assemblies produced were significantly more fragmented (lower $N_{50}$ scores); even when all reads were derived from the PB1 genome sequence, the $N_{50}$ score was only 762 bp (in contrast to the single, full genome contig produced with the read sets generated without mutation). It is interesting to note that while the assemblers could not reconstruct the full genome or longer contigs, virMine still classified contigs as viral and subsequent blastn analyses were able to resolve the origin of the sequence.

## Case study 2: gut microbiomes

Two separate gut microbiome data sets were examined (Table 2). The first includes the sequence data sets that were examined leading to the discovery of the crAssphage genome sequence (97,065 bp) (*Dutilh et al., 2014*): the data set of *Reyes et al. (2010)*. The crAssphage has since been detected in raw sewage and sewage impacted water samples (*Stachler et al., 2017*). Similar to the methods employed in the discovery of the crAssphage, both the sequence data sets of the individual samples and an aggregate of all reads were assembled by virMine using SPAdes (*Bankevich et al., 2012*). Numerous sequences predicted to be viral were identified within the individual samples (727 total) and the aggregate data set (927 total) (File S2). Local blastn analyses identified many of these contigs as representative of transposases and integrases. The abundance of transposase sequences within metagenomic sequences has previously been noted for a variety of environments (*Brazelton & Baross,*

**Figure 3  Coverage of crAssphage by contigs predicted by virMine as viral or unknown.**

*2009*; *Aziz, Breitbart & Edwards, 2010*; *Vigil-Stenman et al., 2017*). We compared the contigs identified as viral to the crAssphage genome sequence (GenBank: JQ995537). 94.88% of the crAssphage genome was represented in 372 contigs identified as viral sequences. Coverage of the crAssphage increases when contigs classified as unknown are considered: 98.32% of the genome is represented in 613 contigs (Fig. 3). Several other complete viral genomes were also identified by virMine including a Gokushovirus and Microvirus exhibiting homology to the sequenced genomes of Gokushovirus WZ-2015a (GenBank: KT264754) and the newly discovered Microviridae sp. isolate ctci6 (GenBank: MH617627). It is worth noting that this Microviridae genome was not included in our viral database and exhibits no significant homology to other records in the current BLAST Nucleotide collection. The second gut microbiota data set was a subset of the fecal samples from 124 European individuals (*Qin et al., 2010*). Most of this data set is bacterial in origin, with only 0.1% predicted by the authors of the study to be of eukaryotic and viral origin. Using virMine we also found that most of the sequences were likely bacterial (File S2). However, we found that the prediction of the study's authors underestimated the viral population; 1.31–38.43% of the assembled contigs were predicted by virMine to be viral in origin. We hypothesize that this discrepancy may be due to prophage sequences. As our previous analysis with the beta version of the software showed, virMine can identify prophage sequences within bacterial genome contigs as well as extrachromosomal viruses (*Garretto et al., 2018*). This underestimate may also be a result of our increased knowledge of viral diversity; the number of viral sequences in GenBank has tripled since the study of *Qin et al. (2010)* was published. The summary of our analysis of the 55 samples from this study are listed in File S2. In total 28,673 and 311,457 contigs were categorized as viral and unknown, respectively.

## Case study 3: urinary viromes

Ten data sets, collected from individuals with urinary tract infections (*Santiago-Rodriguez et al., 2015*), were selected for analysis. In contrast to the gut microbiomes examined in Case Study 2, these samples were prepared such that the majority (if not all) of the sequenced DNA was representative of the viral fraction (*Santiago-Rodriguez et al., 2015*). Exploration of the urinary virome has only recently begun. Of the few viral metagenomic studies of the urinary microbiota (*Santiago-Rodriguez et al., 2015*; *Rani et al., 2016*; *Thannesberger et al., 2017*; *Garretto et al., 2018*; *Miller-Ensminger et al., 2018*; *Moustafa et al., 2018*), most of the identifiable sequences are similar to characterized phage sequences. Nevertheless, the vast majority of contigs exhibit no identifiable homology to sequence databases. As summarized in File S2, each sample consisted of more contigs in the "unknown" category than the "viral" category. We selected the larger contigs (>5,000 bp) that were predicted as viral and queried them via megablast against the nr/nt database online. Table 3 presents

**Table 3   BLAST homology for longer (>5,000 bp) contigs predicted as viral.**

| SRA Run # | BLAST hit | Accession # | Contig length | % ID | % QC |
|---|---|---|---|---|---|
| MGM4568637 | *Cyanothece* sp. PCC 7822 | CP002198 | 14,157 | 73 | 0 |
| | *Choristoneura rosaceana* entomopoxvirus 'L' | HF679133 | 11,424 | 66 | 15 |
| MGM4568639 | *Erlichia canis* strain YZ-1[a] | CP02479 | 12,310 | 73 | 8 |
| | *Burkholderia* sp. MSMB0856 | CP013427 | 5,156 | 71 | 5 |
| MGM4568640 | *Clostridium taeniosporum* strain 1/k | CP017253 | 7,987 | 69 | 2 |
| | *Escherichia* phage YDC107_2 | CP025713 | 5,479 | 96 | 88 |
| | *Enterococcus faecalis* V583[a] | AE016830 | 16,416 | 95 | 95 |
| MGM4568641 | Uncultured Mediterranean phage uvMED | AP013535 | 13,087 | 79 | 1 |
| | *Turicibacter* sp. H121 | CP013476 | 7,825 | 83 | 0 |
| | *Enterococcus faecalis* strain L9[a] | CP018004 | 5,086 | 99 | 100 |
| MGM4568642 | *Choristoneura rosaceana* entomopoxvirus 'L' | HF679133 | 9,301 | 66 | 27 |
| | *Protochlamydia naegleriophila* PNK1 | LN879502 | 5,312 | 83 | 1 |
| MGM4568645 | *Rickettsiales* bacterium Ac37b[a] | CP009217 | 8,302 | 66 | 11 |
| | *Rickettsiales* bacterium Ac37b[a] | CP009217 | 8,215 | 68 | 19 |

**Notes.**
[a]Indicates BLAST homologies to annotated prophage regions.

the results of this search. virMine identified similarities to annotated prophage sequences (indicated by asterisks), extrachromosomal phages, and eukaryotic viral sequences.

## Case study 4: freshwater viromes

Two freshwater viromes were considered. The first includes four samples from the Lake Michigan nearshore waters, collected by our group (*Sible et al., 2015*; *Watkins et al., 2016*). The second includes samples taken from Lough Neagh, the largest freshwater lake in Ireland (*Skvortsov et al., 2016*). The summary statistics for our analysis are included in File S2. Sequences predicted to be viral within the four Lake Michigan data sets were inspected. Hits to known viral sequences varied between samples; in total, sequence homologies were detected to 834 different phage ($n = 425$) and eukaryotic viruses ($n = 409$). Figure 4 illustrates the species, predominantly phages, with the most hits. From the Lough Neagh data set, nine contigs were identified by virMine as viral and had a length greater than 40 Kbp. In the study introducing this data set (*Skvortsov et al., 2016*), only five contigs were produced meeting this length threshold. (The IDBA-UD assembler (*Peng et al., 2012*) was used in the original analysis of this data set (*Skvortsov et al., 2016*).) Each contig was submitted to RAST (*Aziz et al., 2008*) for annotation and each was found to contain phage-related genes (File S3), suggesting that the contigs represented complete or partial phage genomes. We next queried each contig against the nr/nt database via blastn identifying only modest sequence homology to bacterial, phage, and uncultured viral isolate sequences (Table 4). These contigs thus represent likely novel viral sequences.

## virMine performance

To assess the performance of virMine, the freshwater data sets were also examined using the viral sequence identification tool VirSorter (v. 1.0.3) (*Roux et al., 2015*). For all five
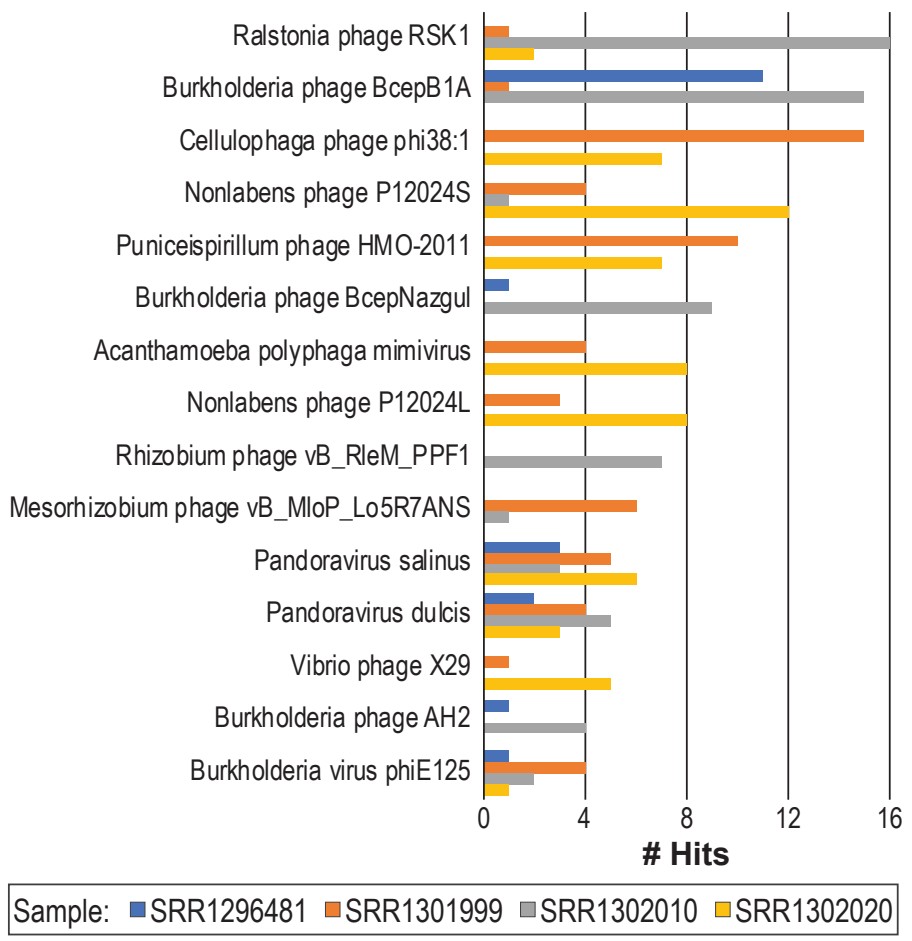

**Figure 4** **Viral species most frequently detected within the Lake Michigan data sets.**

data sets, we found that very few contigs were predicted as viral by both tools. For instance, in the Lough Neagh data set, VirSorter only identified (a category 2 prediction) one of the nine virMine contigs (length > 40 Kbp). This prompted our manual inspection of these results. Herein we present the results for one of the samples from Lake Michigan (SRA accession number SRR1296481), representative of what we found in all sets. virMine predicted 60 of the 1,518 assembled contigs as viral. VirSorter predicted only 20 viral sequences (two category 1; five category 2; six category 3; no category 4; four category 5; and three category 6). Only two sequences were predicted by both tools. As virMine was designed for identifying viral contigs and VirSorter was designed to identify both viral contigs (categories 1–3) and prophages (categories 4–6), it is not surprising that both contigs detected by the two tools were VirSorter category 2 sequences. (While virMine can identify prophages, as was shown previously (*Garretto et al., 2018*), it will not identify prophages within large bacterial contigs.) BLAST queries to the nr/nt database of the sequences uniquely identified by virMine and VirSorter are listed in File S3; many of these predicted sequences exhibited homology to bacterial RNAs (rRNA and tRNA). Only four

**Table 4  Viral genome sequences identified by virMine from the Lough Neagh virome (*Skvortsov et al., 2016*).**

| Contig | Length | # CDS | BLAST hit | Accession # | % QC | % ID | Isolation source |
|---|---|---|---|---|---|---|---|
| contig_11 | 46,867 | 71 | *Chromobacterium rhizoyzae* strain JP2-74 | CP031968.1 | 1 | 80 | Rhizosphere |
| contig_12 | 46,702 | 74 | Uncultured marine virus isolate CBSM-242 | FJ640348.1 | 0 | 83 | Chesapeake Bay sediment |
| contig_13 | 46,245 | 60 | Bacteriophage 11b | AJ842011.2 | 1 | 68 | Arctic sea ice |
| contig_17 | 40,578 | 56 | *Methylobacterium brachiatum* strain TX0642 | CP033231.1 | 6 | 67 | Automobile air-conditioning evaporator |
| contig_18 | 40,568 | 61 | *Blastochloris* sp. GI | AP018907.1 | 0 | 72 | Soda dam hot springs |
| contig_2[a] | 70,520 | 92 | Uncultured virus YBW_Contig_50752 | KU756933.1 | 1 | 72 | North Sea Surface Water Virome |
| contig_5 | 56,143 | 55 | Uncultured virus SERC 372681 | KU595468.1 | 2 | 73 | Rhode River surface water |
| contig_6 | 55,961 | 75 | *Polynucleobacter asymbioticus* strain MWH-RechtKol4 | CP015017.1 | 1 | 71 | freshwater |
| contig_7 | 55,939 | 77 | Uncultured virus SERC Contig 695464 | KU971113.1 | 0 | 76 | Rhode River surface water |

**Notes.**
[a]Contig also predicted as viral by VirSorter (*Roux et al., 2015*).

additional sequences (two predicted by virMine and two predicted by VirSorter) exhibited homology to genes/sequences annotated as phage.

Our comparison of virMine to VirSorter highlights the importance of manual inspection of results. In contrast to VirSorter and, e.g., VirFinder, virMine not only predicts viral sequences but also reports the blast results of these sequences. This aids in the manual inspection of the virMine predictions. It is important to note that our comparison here, however, is not entirely an equivalent assessment: VirSorter relies on a different sequence database than virMine. As described in *Roux et al. (2015)*, two reference databases are used by VirSorter. These databases have been updated to version 2 since the time of its publication, and details regarding this update are not readily available. In fact, the viral databases used by existing tools varies greatly. VirSorter and MARVEL restrict their viral database to phages, all phages and dsDNA phages from the *Caudovirales* order, respectively. However, virMine includes all viral sequences—phages as well as eukaryotic viruses. As shown in Fig. 4, a number of hits to eukaryotic viruses were identified within the Lake Michigan data sets. While VirusSeeker's database is not restricted to phage sequences, as it too contains eukaryotic viral sequences, it is a curated database (last updated August 2016). Currently, MetaPhinder's and MetaVir's databases are also out of date; both were last updated in 2017. virMine's database is entirely controlled by the user and can include all data currently available. Just as virMine allows the user to create their own custom databases, so too does FastViromeExplorer. FastViromeExplorer requires the user to format files for use. In contrast, virMine only necessitates a multi-fasta file which can easily be retrieved from publicly available databases like NCBI and IMG/VR or via user-specific queries of public sequence repositories.

## CONCLUSIONS

As highlighted in the recent report of the International Committee on Taxonomy of Viruses (ICTV) Executive Committee, genomes identified from metagenomic data will vastly expand our catalog of viral diversity (*Simmonds et al., 2017*). Within just the past two years, there has been an explosive growth of the number of uncultivated viral genomes identified within metagenomic data (*Roux et al., 2018*). Our analysis of complex communities has uncovered numerous novel viral genomes. virMine is capable of identifying both prophages in contigs and viral sequences. In contrast to other tools that rely solely on viral sequence availability, virMine makes use of a far larger, more comprehensive data set—non-viral sequences. Furthermore, the entire process from raw sequence quality control through analysis is packaged into a single tool providing a "consensus" solution for viral genome discovery (*Dutilh et al., 2017*). Manual inspection of virMine results can thus lead to the identification of viral sequences resembling known viruses as well as novel viral strains. As exemplified here, virMine can be used to identify viruses in any niche and thus further our understanding of this vast reservoir of genetic diversity.

## ACKNOWLEDGEMENTS

The authors would like to thank Ms. Ally Miley for conversations during the development of this tool and Dr. Jason Shapiro for his feedback on earlier versions of the manuscript.

### Funding

This work was supported by the National Science Foundation (grant number 1149387) to Catherine Putonti. Andrea Garretto was supported by Loyola University Chicago's Carbon Research Fellowship and the CRA-W's CREU program. The funders had no role in study design, data collection and analysis, decision to publish, or preparation of the manuscript.

### Grant Disclosures

The following grant information was disclosed by the authors:
National Science Foundation: 1149387.
Loyola University Chicago's Carbon Research Fellowship.
CRA-W's CREU program.

### Competing Interests

The authors declare there are no competing interests.

### Author Contributions

- Andrea Garretto performed the experiments, analyzed the data, prepared figures and/or tables, authored or reviewed drafts of the paper, approved the final draft.
- Thomas Hatzopoulos performed the experiments, analyzed the data, approved the final draft.
- Catherine Putonti conceived and designed the experiments, analyzed the data, prepared figures and/or tables, authored or reviewed drafts of the paper, approved the final draft.

## Data Availability

Data is available at GitHub: https://github.com/thatzopoulos/virMine.

## Supplemental Information

Supplemental information for this article can be found online at http://dx.doi.org/10.7717/peerj.6695#supplemental-information.

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
