# Peer review of "virMine: automated detection of viral sequences from complex metagenomic samples"

_PeerJ, doi:10.7717/peerj.6695_

## Round 0.1 · original submission · Minor Revisions

The manuscript is well written and provided best use cases for the software. The end result demonstrates that the virMine package has its place as a valuable tool in screening for viral sequences. There is brief mention of some other tools; however, perhaps a deeper discussion of the similarities and contrasting differences might be expanded upon. The reviewers do mention some of these concerns and these should be addressed to appease their comments. Another aspect to consider might be to address the potential use of this pipeline with the Galaxy project scientific workflow environment (also developed in python). Regarding some of the test data-sets might you consider developing some sort of Docker environment to allow testing, or perhaps utility; updated databases could be made available through mounted volumes. I’m not sure how detailed this would be addressed, but how would this tool work in transposon-rich environments. Please address these concerns to the best of your ability, and we can move this forward. Thank you for your contribution.

Reviewer 1 ·

Basic reporting

The paper describes the software virMine and its application to metagenomics. The work is well written. There is a comprehensive introduction to the subject of viral metagenomics and software that can be used to identify viral contigs within metagenomes. There are two pieces of software that should be included which aren’t, these are MARVEL and DeepVir.
The software takes a different approach to identify viral contigs compared to other software, using the extensive datasets that are not viral to classify viral contigs, which is an interesting approach. The software is an all in one approach, taking raw reads and running through QC, assembly, gene prediction and classification. This in its self is a useful tool.

It would help to have the tools that can be used, to be labelled in Figure 1, allowing an nonexpert to see what tools are used at each step

Experimental design

The paper then goes onto described a number of test cases
The first test case uses a phage and bacterial genome sequence, whereby they demonstrate that virMine can assemble these genomes and identify the phage contigs. Thereby showing as a proof of principle the pipeline works on a simple synthetic system.

Case study 2 then focuses on using the tools on metagenomics datasets that have previously been analysed. Here new viruses were discovered that had not previously been reported in the original studies, which is an interesting result. However, if this a result of the approach taken in this study or simply better databases –the original analyses are between 5 and 8 years old. With significant increases in database sizes since the original publication. It is not clear if these viruses would be identified if the original analysis was carried out with updated databases. Furthermore, it is not clear if virMine is a better tool at identifying viral contigs than any other tool, such as DeepVir, VirSorter etc

Case study 3, the tools are used to analyse urinary viromes, where it identifies numerous contigs that are viral. Again it is not clear if in this example the tool is better than other methods for identifying viral contigs .

Case study 4 uses the tool to analysis viromes from a freshwater lake, which has previously been produced by the same group. They also compare the analysis to results produced by VirSorter and highlight some differences.

At the end of this analysis, it is not clear if the all in one pipeline is better than any tools that are currently available for virome analysis. In terms of a one-stop shop, then the pipeline is easy to use as it takes reads and outputs contigs that maybe viral.
Undoubtedly, the initial part of the pipeline is useful to be able to QC and assemble reads. However, there is no evidence that the sorting approach is better than current tools such as VirSorter DeepVir etc at identifying contigs or an equal to them. There is also do data presented on the speed and accuracy of the tool. Without this information it hard to make a decision on whether to use this tool rather than any other tool
I would recommend a more thorough comparison of the viral comparison tool against other leading tools such as VirSoter, VirFinder. This could be done using a similar approach as was carried out by Ren J et al 2017 or provide data on how accurate the tools is on more complex synthetic datasets to allow accuracy, speed to be compared to other tools.
Clearly, a large amount of work has gone into this tool and there is much to be commended in terms of how the paper is clearly written and the provision of docker images. However, without this crucial data on how the tool performs against other tools, it is not possible to assess its usefulness.

Minor points
Line 76- The use of “bin” is ambiguous in this sentence. The Virfinder tools gives each contig a probability of being viral or not, it is not binned as such. Please clarify what is meant
Line 83- MARVEL should be included in this list of software

Line 133
The premise of the paper makes much of the comparison against databases and the approach is innovative. However, much will depend on the databases used. Looking at the script that creates the viral database in https://github.com/thatzopoulos/virMine/blob/master/virmine_make_dbs.py
Line 10- the script downloads the NCBI viral database from the FTP site. Whilst a good starting point the database only includes a small number of complete bacteriophage genomes that are available in Genbank, missing ~ 8000 phage genomes
Having a more comprehensive set of phage genomes as the starting point would improve the use of the tool – The authors do state in the text this can be manually curated set (Line 141-143) that would allow users to add in more phage genomes. Based on the GitHub repository it is not immediately clear how this can be done. Instructions on how to do this would significantly improve the usefulness of this software.
Line 168- as stated above the RefSeq database excludes 8000 complete phage genomes that are available
Line 249 –it is not clear when the database was created. But the sequence of the highlighted phage: MH617627 is a clear example of why a more up to date viral database is needed
Line 257 – whilst there is a clear underestimation of the vital data in the Qin dataset, it is not clear if this is due to better performance of virMine or simply increased database size in 2018 than in 2010. The way the analysis is carried out do not allow this to be determined

Validity of the findings

The data does support that conclusions, that the tool can be used to identify viral contigs from raw reads in a single pipeline. However, there are other tools that do this - not in single pipeline admittedly. I would argue the identified viral contigs should be predicted as accurately as other tools can ,which currently cannot be assessed

·

Basic reporting

Reporting is clear. Context was developed early in Intro, and the figures are well presented.

Experimental design

The experiments were well designed with sufficient details provided.

Validity of the findings

Conclusions are supported by the results.

Additional comments

Work reported:
virMine is presented as a new software to identify viral genomes by omitting non-viral sequences from raw reads. Currently available tools such as VIP, VirFinder, Phage Eco-Locator, VIROME, MetaVir, VirSorter, MetaPhinder, VirusSeeker and FastViromeExplorer rely upon homologies to known viral sequences, and generally require manual curation. virMine was used as a single software to distinguish viral sequences in four case studies presented. virMine performed well with synthetic data set, it reconstructed viral genome when more than half of the set included specific viral sequences. The software identified two transposases as viral sequences because these were categorized as mobilomes (COG category X) and were excluded from non-viral set. Second case study was performed with gut data sets. In the first, virMine was able to assemble crAssphage, and identified transposases and integrases, and other complete viral genomes. In the second gut data set, expected to be mainly bacterial, virMine found higher %-age of viral sequences than previously reported. The authors suggest these to be prophage sequences. In the third case study urinary viromes from individuals with urinary tract infections were examined. Several unknown category sequences were found with similarities to viral sequences. For case study 4, two freshwater lake samples were examined. VirSorter was used for comparison with virMine. Manual curation was performed with both methods. virMine predicted higher number of sequences as viral (60 vs. 20), however, both software assembled the same two contig sequences belonging to VirSorter category 2. Both software picked sequences that belonged to bacterial rRNA and tRNA, and identified two each (different sequences) with similarity to phages. Manual curation was necessary for both tools. With the last dataset, virMine found 9 contigs likely to represent new viral sequences. VirSorter identified one of the contigs and previous work had picked 5 contigs. Overall, virMine performed better in identifying more unknown contigs.

Critical comment:
1. virMine utilizes a clever approach to subtract known (non-viral) sequences. As demonstrated by the authors, this approach works well with datasets with higher representation of viral sequences. When the datasets are more complex and have lower representation of viral sequences, this approach requires manual curation, particularly to confirm that the identified sequences are indeed of viral origin. Several statements should be toned down about virMine as a fully automated software tool. For example “virMine automates sequence read quality control, assembly, and annotation” and “virMine is a single tool to perform raw read quality control, assembly, and analysis”. The data shows virMine is not fully automated, it requires an essential manual curation and interpretation step. This should be made clear throughout the manuscript. It does not take anything away from the ability of virMine to distinguish between viral and non-viral sequences, it is important to set the right expectation early on. This reviewer was disappointed that in the end the software did not perform fully as was promised earlier in the text.
2. As expected from the approach, virMine identified sequences that were not well identified or annotated in the databases, including transposases, integrases, prophage sequences, and even bacterial rRNA and tRNA sequences. virMine uses SPAdes or metaSPAdes for assembly, contigs are queried using blastx. Modified GLIMMER script is used for ORF prediction and compared to viral and non-viral datasets. It might be of great advantage to integrate one more step, borrowing from other software such as VIROME and VirSorter, to search for recognizable homologies to known viral sequences after the non-viral sequences have been differentiated. This step will add value in increasing the confidence of viral origin prior to the manual step, and could be useful for the end users to categorize their database searches. The authors should comment if they intend to integrate an additional “vral recognition” step, or if not, why it may not be necessary. I believe this will be of interest to the readers.
3. All in all, virMine is a good approach for identifying new viral sequences. However, only one software “VirSorter” was compared against virMine and only for case study 4. The comparison showed that while virMine predicted higher number of viral sequence, both software performed similarly in assembling contig sequences and both required manual curation. It should be of great interest to the authors to test the performance of virMine with VirSorter with other case studies, such as the second gut database and the infected urinary viromes. In addition, performance of other software should be compared, such as VirFinder to compare a broader set of approaches (nucleotide usage based approach). These direct comparisons are likely to illuminate the utility of virMine further, and will add significantly to this manuscript.

Minor comments:
Noticed (but not noted) minor typographical errors. The manuscript should be screened carefully for extra brackets and other minor errors.

---

## Round 0.2 · accepted · Accept

This latest version of the manuscript reads very well, and it appears you have used the comments from reviewers in a way which strengthens the utility of your software package. With the diverse sample examples, public availability and test-ability, I feel that virMine should be well-received and should gain popular usage with the ever expanding field of genome surveys. I think it is ready as is, as I have no suggested edits. Congratulations on your manuscript; please consider it accepted for publication.

#